# Interaction of a Polyarginine Peptide with Membranes of Different Mechanical Properties

**DOI:** 10.3390/biom9100625

**Published:** 2019-10-18

**Authors:** Matías A. Crosio, Matías A. Via, Candelaria I. Cámara, Agustin Mangiarotti, Mario G. Del Pópolo, Natalia Wilke

**Affiliations:** 1Facultad de Ciencias Químicas, Departamento de Química Biológica Ranwel Caputto, Universidad Nacional de Córdoba, X5000HUA Córdoba, Argentina; matias.crosio@gmail.com (M.A.C.); amangiarotti@unc.edu.ar (A.M.); 2Centro de Investigaciones en Química Biológica de Córdoba (CIQUIBIC), CONICET, Universidad Nacional de Córdoba, X5000HUA Córdoba, Argentina; 3ICB-CONICET & Facultad de Ciencias Exactas y Naturales, Universidad Nacional de Cuyo, M5502JMA Mendoza, Argentina; matysvia@gmail.com (M.A.V.); mdelpopolo@fcen.uncu.edu.ar (M.G.D.P.); 4Facultad de Ciencias Químicas, Departamento de Fisicoquímica, Universidad Nacional de Cordoba, X5000HUA Córdoba, Argentina; candecamara@gmail.com; 5Consejo Nacional de Investigaciones Científicas y Tecnicas, CONICET, Instituto de Investigaciones en Fisicoquímica de Córdoba, INFIQC, X5000HUA Córdoba, Argentina; 6Instituto de Investigación Médica Mercedes y Martín Ferreyra (INIMEC-CONICET-Universidad Nacional de Córdoba), X5016NST Córdoba, Argentina

**Keywords:** cell penetration peptides, liquid-ordered phase, membrane heterogeneities, membrane rheology

## Abstract

The membrane translocation efficiency of cell penetrating peptides (CPPs) has been largely studied, and poly-arginines have been highlighted as particularly active CPPs, especially upon negatively charged membranes. Here we inquire about the influence of membrane mechanical properties in poly-arginine adsorption, penetration and translocation, as well as the subsequent effect on the host membrane. For this, we selected anionic membranes exhibiting different rigidity and fluidity, and exposed them to the nona-arginine KR_9_C. Three different membrane compositions were investigated, all of them having 50% of the anionic lipid 1,2-dioleoyl-sn-glycero-3-phospho-(1’-rac-glycerol) (DOPG), thus, ensuring a high affinity of the peptide for membrane surfaces. The remaining 50% was a saturated PC (1,2-dipalmitoyl-sn-glycero-3-phosphocholine, DPPC), an unsaturated PC (1,2-dioleoyl-sn-glycero-3-phosphocholine, DOPC) or a mixture of DOPC with cholesterol. Peptide-membrane interactions were studied using four complementary models for membranes: Langmuir monolayers, Large Unilamellar Vesicles, Black Lipid Membranes and Giant Unilamellar Vesicles. The patterns of interaction of KR_9_C varied within the different membrane compositions. The peptide strongly adsorbed on membranes with cholesterol, but did not incorporate or translocate them. KR_9_C stabilized phase segregation in DPPC/DOPG films and promoted vesicle rupture. DOPC/DOPG appeared like the better host for peptide translocation: KR_9_C adsorbed, inserted and translocated these membranes without breaking them, despite softening was observed.

## 1. Introduction

Cellular membranes regulate lateral diffusion of the lipids and proteins associated to them, compartmentalization and permeability. It has been shown that organisms adapt the lipid composition of their membranes in order to maintain them mainly in a fluid state [1]. Notwithstanding, several studies performed in plasma and internal membranes point to the existence of regions with different compositions, which leads to local differences in mechanical properties. While the presence of membrane proteins has been related to solid docks [2], sterols are accepted as liquid-ordered phase inducers [3]. Thus, the current model for membranes is a patchwork-like surface, with the different regions being highly variable both, in size and in time [3].

Soluble molecules that interact with membranes, such as peptides may have affinities that depend on membrane composition and rheology. As a consequence of the patchwork-like character of the membrane, regions with a broad spectrum of properties are available for the interaction with soluble peptides. Thus, it is important to know to what extent the peptide-membrane interactions depend on membrane rheology, and also how the membrane mechanical properties change after the interaction has taken place.

Cationic peptides have been widely investigated during the past years, and among other classifications, they have been sorted into two groups: Antimicrobial peptides (AMPs) and cell penetrating peptides (CPPs). The main difference between these families is the peptide-lipid affinity, which seems to be higher for the antimicrobial peptides. Besides, at low peptide concentrations AMPs promote membrane lysis, while CPPs are able to cross membranes without largely affecting their properties [4,5]. As such, CPPs have become an invaluable tool in cell biology, allowing to introduce CPP-functionalized cargoes into cells [6,7,8].

Since the discovery of CPPs [9,10], a significant amount of information has been obtained related to their cell-permeability. Peptide amphipathicity, length and amino acid sequence have been pointed out as important factors [7,11]. Basic residues enhance the efficiency of peptides to cross membranes. Related to this, an extraordinary ability to passively penetrate cellular membranes has been reported for arginine-rich polypeptides [12,13], the observation that coined the term “Arginine-magic” [11,14,15].

Lesser information is available in relation to how membrane properties affect the CPP action. Main observations so far include that CPP translocation is fostered by the presence of fatty acids [13,16], as well as by lipids that form hexagonal II phase [17,18], whilst cholesterol (CHOL) hinders CPP translocation [19,20]. 

With all these in mind, our aim was to inquire about how the affinity and consequent effects of the interaction between a CPP-like peptide and membranes are affected by the membrane mechanical properties. This kind of studies using lipid membranes is a first step towards understanding the complex regulation of the interaction of membranes with soluble molecules by the membrane rheology, as well as the subsequent effect of the interaction on membrane mechanical properties. We studied the interaction and the associated effects of a polyarginine with binary membranes. Highly anionic membranes were chosen (with 50% of 1,2-dioleoyl-sn-glycero-3-phospho-(1’-rac-glycerol), DOPG) to assure lipid-peptide interaction with all membrane compositions. The other 50% was a saturated lipid (1,2-dipalmitoyl-sn-glycero-3-phosphocholine, DPPC), an unsaturated lipid (1,2-dioleoyl-sn-glycero-3-phosphocholine, DOPC) or a mixture of DOPC and CHOL. In this manner, we studied monolayers in liquid-condensed/liquid-expanded or bilayers in gel/liquid-disordered phase coexistence (DPPC/DOPG) [21], liquid-expanded monolayers or liquid-disorder bilayers (DOPC/DOPG), and mixtures with CHOL, which induces more compact, but still fluid DOPC membranes [22,23,24].

The selected peptide was a nona-arginine, since this length was found optimal regarding translocation efficiency across lipid membranes [25]. We also included in the sequence a lysine and cysteine residues, forming the peptide KR_9_C, with the aim of broadening its possible applications, since these aminoacids allow to bind the polyarginine to proteins (lysine residue) and to gold surfaces (cysteine residue).

We studied peptide insertion into Langmuir monolayers, adsorption on large and giant unilamellar vesicles (LUVs and GUVs), and translocation through black lipid membranes (BLMs). Our results indicate that membrane rheology is an important factor for peptide adsorption, incorporation and subsequent translocation. KR_9_C peptide actively interacted with DPPC/DOPG: It adsorbed, incorporated and crossed these membranes. However, phase segregation and membrane lysis were observed, pointing to rigid, but fragile membranes. On the other hand, the peptide did not incorporate into membranes with CHOL, but it remained at their surfaces. Finally, favorable adsorption, penetration and translocation were observed when this peptide was added to membranes formed by the two unsaturated lipids. We observed that after CPP addition to DOPC/DOPG membranes, vesicle shape fluctuations increased and the membrane resistance upon curvature changes became weaker, pointing to a softening of the lipid bilayer. 

## 2. Materials and Methods 

KR_9_C and 5-FAM-KR_9_C ≥ 95% purity was purchased from Innovagen (Lund, Sweden). NaCl, polyvinyl alcohol (PVA), n-decane, albumin (from bovine serum) and chloroform were obtained from Sigma-Aldrich (St. Louis, MO, USA). Cholesterol (CHOL), 1,2-dioleoyl-sn-glycero-3-phosphocholine (DOPC), 1,2-dipalmitoyl-sn-glycero-3-phosphocholine (DPPC), 1,2-dioleoyl-sn-glycero-3-phospho-(1’-rac-glycerol) sodium salt (DOPG), 1,2-distearoyl-sn-glycero-3-phosphoethanolamine-N-[biotinyl(polyethylene glycol)-2000] ammonium salt (DSPE-PEG(2000) Biotin), and the lipophilic fluorescent probe L-a-phosphatidylethanolamine-N-(lissamine rhodamine B sulfonyl) (ammonium salt) (egg-transphosphatidylated chicken) (Rho-PE) were obtained from Avanti Polar Lipids (Alabaster, AL, USA). The chemical structure of the lipids is depicted in Appendix A (Appendix A). Streptavidin-coated micro-beads (3 μm) were purchased from Bangs Laboratories Inc. (Fishers, IN, USA), Lipids were dissolved in 2:1 ratio chloroform:methanol (1 nmol/mL). All the solvents and chemicals used were of the highest commercial purity available. Peptide and NaCl solutions were prepared using deionized water with a resistivity of 18 MΩ cm, filtered with an Osmoion system (Apema, BA, Quilmes, Argentina).

### 2.1. Methods 

#### 2.1.1. Monolayers at the Air-Water Interface

Compression isotherms and monolayer insertion experiments were performed in a NIMA 102M (NIMA Technology, Coventry, UK) Langmuir balance, using the Wilhelmy method with a platinum plate. The lipid monolayers were formed by spreading the lipids on a NaCl 150 mM solution prepared at the water pH (about six, due to CO_2_ dissolution), and compressed at 1 × 10^−2^ nm^2^ molecule^−1^ min^−1^. Insertion studies were performed on a homebuilt trough with 1 mL capacity. The lipid solution was spread, drop by drop, onto the subphase until reaching the desired initial surface pressure (π_0_). Following solvent evaporation (5 min), an aliquot of the peptide solution was injected in the subphase through a hole in the wall of the trough. The final concentration of the peptide was 10µM, as this concentration was previously found to be the minimum necessary to saturate the incorporation of KR_9_C into perfluorotetradecanoic acid monolayers [26]. Time evolution of the surface pressure was registered after the addition of the peptide and its final value, π_f_, was determined from the asymptote of the curve. The change in surface pressure was defined as ∆π = π_f_ − π_0_. All the experiments were performed at 25 ± 1 °C. Langmuir monolayers were observed with a direct microscope (Axioplan, Carl Zeiss, Oberkochen, Germany) with 20× objective (air immersion) and a CCD video camera Axio CamHRc (Carl Zeiss, Oberkochen, Germany). In these experiments, 0.5 mole% of Rho-PE was included in the lipid mixtures. 

#### 2.1.2. Large Unilamellar Vesicles (LUVs), Hydrodynamic Size and Z-Potential Measurements

A uniform lipid film was formed on the wall of a glass tube by solvent evaporation under an N_2_ flow from a chloroform:methanol lipid solution. Final traces of solvent were removed by incubating the lipids in a high-vacuum chamber for 1 h. Then, the lipids were resuspended in the aqueous solution of NaCl 1 mM, to a final concentration of 0.3 mM. The suspension was incubated in a 50 °C water bath for 30 s, after that, it was vortexed and subjected to cold incubation for 30 s. This procedure was repeated nine times. The resulting multilamellar vesicles were extruded 21-times through a 100 nm pore filter (Avanti) at 50 °C. 

For the z potential and dynamic light scattering (DLS) measurements, the 0.3 mM lipid dispersion was diluted to a final concentration of 0.12 mM. Then, an appropriate volume of a stock solution of the peptide was added to 100 µL of this dilution to achieve the desired peptide to lipid molar ratio (P/L), and incubated for 1 h. The z-potential determinations were performed by means of Henry’s equation, subjecting aliquots of 100 µL of liposomes and peptide to an electric field and determining electrophoretic mobility with Z-sizer SZ-100-Z equipment (Horiba, Ltd., Kyoto, Japan).

In order to calculate the P/L ratio, the amount of phospholipid was quantified using the Bartlett’s method [27]. This is, phospholipids were digested, and inorganic phosphorus was quantified colorimetrically. Absorbance was measured with a Shimadzu UV-visible Spectrophotometer (Biospec-mini, Chiyoda-ku, Tokyo, Japan) at λ = 830 nm.

#### 2.1.3. Giant Unilamellar Vesicles (GUVs)

A 100 μL aliquot of a 5% (W/W) solution of PVA in water was spread on the glass base of a Petri dish, and dried for 1 h at 50 °C. Ten microliters of a 2.5 μg/mL lipid solution doped with 0.5 mole% of the fluorescent probe Rho-PE were spread onto the PVA film, subjected to vacuum for 1 h for solvent evaporation, and hydrated with 300mM sucrose in 15 mM NaCl. GUV formation was corroborated after 1 h with optical microscopy using a confocal fluorescence microscope (Confocal Zeiss FV1000) with a 60× objective (immersion in oil). For observations, an aliquot of the GUV suspension was transferred to the observation chamber containing 300 mM glucose in 15 mM NaCl. Previously, the observation chamber was treated with a 10% (w/v) of albumin solution, which prevented the rupture of GUVs on the glass slide. Osmolarity parity between sucrose and glucose solutions was tested with an automatic micro-osmometer OM-806 (Vogel, Germany). 

For peptide adsorption determinations, 1.2 or 3 μL of the fluorescently labeled peptide 5-FAM-KR_9_C (200 μM) were gently added to 240 μL of the GUV’s suspension in the observation chamber, and the increase in fluorescence intensity at the membrane surface, and in the GUV’s surroundings was followed over time. For this, we determined the grey level inside 3–5 ROIs close to each GUV and at the GUV’s rim along with a stack of images and plotted this data as a function of time, starting from the time where the fluorescence increased, as exemplified in Appendix A (Appendix A). 

Shape fluctuations were determined for the GUVs composed of DOPC/DOPG, registering 15−20 individual GUVs through time (100 frames, 38 frames/min) as detailed in ref. [28]. Briefly, for each vesicle, the value of the aspect ratio (AR = longer axis length/minor axis length) was determined along with the videos, and the standard deviation of the AR values (SDAR) was statistically analyzed. Higher values of SDAR represented vesicles with more fluctuating shapes than lower values. Each vesicle was classified as “fluctuating” or “non-fluctuating” depending on the value of SDAR. For this, a reference value of SDAR was defined as the P95th value (the value below which 95% of the data are found) of SDAR in the control condition (without peptide). With this reference value, each GUV was considered “fluctuating” if the SDAR value was higher than the reference one or else, “non-fluctuating”. The percent of fluctuating GUVs was calculated as the number of fluctuating GUVs divided by the total number of analyzed GUVs. This procedure was performed in two independent sets of experiments and the averages ± SE are shown. A set of data is shown in Appendix A (Appendix A). 

For the nano-tube retraction experiments, the lipid solutions contained 1 × 10^−4^ mole% of a biotinylated lipid (DSPE-PEG(2000) Biotin). An aliquot of the GUV suspension was transferred to a home-made observation chamber containing 300 mM glucose in 15 mM NaCl. Previously, the observation chamber was treated with an avidin solution in order to attach GUVs and prevent their motion. Nano-tubes were formed by attaching a streptavidin-coated micro-sphere to the membrane, and moving the bead with an optical tweezer. Beads were previously cleaned by ten consecutive cycles of centrifugation, pellet separation and resuspension in clean water. 

#### 2.1.4. Optical Tweezer Setup

The optical trap was formed with the Gaussian beam from a 3W power ND:YVO_4_ laser (Spectra-Physics, Santa Clara, CA, USA) focused through a 100× objective (water immersion, NA = 1). The tweezer was mounted in an Axiover 200 inverted microscope (Zeiss), in a similar manner to that described by Lee et al. [29], and the experiments were registered with a fast CCD video camera (IxonEM+ model DU-897, Andor Technology, Abingdon, Oxfordshire, England). The original microscope stage was replaced by a motorized one (BeijingWinner Optical Ins. CO., LTD, Langfang, Hebei, China), in order to move the GUV attached to the cover-slip in a controlled manner. The trapped bead was then separated from the GUV’s surface, thus, generating the membrane nano-tube. The trap was turned off with a shutter after a 20–30 µm length nano-tube was generated, and the retraction process was recorded. Three to nine nanotubes were generated from each GUV before and after addition of 2 μL of the peptide (final concentration 0.25 μM). Retraction kinetics was recorded and analyzed, and the data shown correspond to the average ± SE obtained from two different GUVs in two independent experiments. 

#### 2.1.5. Bilayer Lipid Membrane (BLM)

Black lipid membranes (BLMs) were formed in a polysulfone cuvette with a 150µm-aperture inserted into a hold in a black Derlin chamber (model CF13A-150 and BCH-M13, Warner Instruments Inc., Hamden, CT, USA). The electrical recordings were performed with a potentiostat/galvanostat Autolab PGSTAT30, equipped with impedance (FRA) and low current (ECD) modules. Current and potential data were acquired using NOVA 2.1.2 software booth Metrohm Autolab B. V. The electrochemical cell was formed immersing in the cis compartment a Pt wire (counter electrode) and Ag^+^/AgCl/Cl^−^ (reference electrode) and in the trans compartment a Pt wire (working electrode), as previously performed by Khan et al. [30]. 

An aliquot of the lipid mixtures in chloroform was dried under N_2_ stream and put under vacuum for 1 h for solvent removal. After this, lipids were resuspended in the corresponding volume of n-decane to a final concentration of 25 mg/mL. BLM was formed by pre-coating the hole with lipid/n-decane solution from the cis side using a glass sphere. After the lipid solution was dried, both compartments were filled with 1mL NaCl 150 mM, previously filtered through a 0.2 µm nylon filter. Then, the cis side of the hole was painted once more with lipid/n-decane solution [31]. 

Bilayer formation was tested by impedance spectroscopy 30 min after painting. The presence of the BLM leads to an increase in the resistance from MΩ (in the absence of the bilayer) to GΩ range (in the presence of the bilayer), together with a capacitance value between 0.4–0.9 µF/cm^2^. Resistance and Capacitance values were obtained by fitting the experimental data with equivalent electrical models for planar bilayer [30,32]. The used equivalent circuit, together with representative plots, is shown in Appendix A (Appendix A).

After corroborating the presence of a stable bilayer, chronoamperometry measurements were performed. The bilayer was left for 600 s at 0V with respect to the open circuit potential (OCP), and after this, a +150 mV (with respect to OCP) was applied. After other 600 s, 3 µL of 366 µM peptide was added to the cis compartment reaching a final concentration of 1 µM. The current was registered continuously, and the change in current after peptide addition was taken as a measure of membrane permeability and peptide translocation. See representative recordings in Appendix A (Appendix A).

For each membrane composition, 5 or 6 independent experiments were performed, and the results shown are the average ± SD. Membrane breakdown, due to the applied potential was not observed in our experiments, and the level of noise in the current recordings remained unchanged at 150 mV compared to 0V. We, thus, assumed that the applied potential did not affect the state of the membrane, as previously reported in DOPC/DPPC bilayers [33]. The chamber and cup were cleaned following the protocol described by Warner instruments, washing several times with a sodium phosphate tribasic solution (50mM), HCl 0.1 % *w/v* and ultra-pure water.

## 3. Results

### 3.1. KR_9_C Recruits DOPG and Does Not Incorporate into Membranes with Cholesterol

With the aim of studying the interaction of CPPs with membranes of different mechanical properties, we studied the incorporation of KR_9_C into monolayers that contained DOPG and DOPC, DPPC or DOPC/CHOL (see chemical structures in Appendix A). A high percentage of the anionic lipid was included (50%) in order to assure the lipid-peptide attraction. All experiments were performed using the following membrane compositions: DPPC/DOPG 1:1 (liquid-condensed/liquid-expanded-see below- or gel/liquid-disordered phase coexistence [21]), DOPC/DOPG 1:1 (liquid-expanded or liquid-disordered membranes) or DOPC/CHOL/DOPG 3:2:5 (more compact than in the absence of CHOL, but still fluid membranes [22]).

The incorporation of KR_9_C was also studied in monolayers composed of the pure lipids. Monolayers were prepared at the desired surface pressure (π_o_), and after equilibration, the peptide was injected into the subphase to a final concentration of 10 µM. In the absence of monolayer, the surface tension remained constant after peptide addition. In the presence of monolayers, the surface pressure was recorded over time. The total pressure changes (Δπ = π_f_ − π_o_) induced by the peptide are summarized in Figure 1a for all monolayer compositions except for CHOL and DOPC/CHOL/DOPG 3:2:5, where no change in surface tension was detected. This indicates that KR_9_C did not incorporate into these monolayers, although the interaction of the peptide with the polar head-groups of the lipids cannot be ruled out. 

For all other compositions, an increase in surface pressure was observed, with the highest Δπ values being for DOPG and DOPC/DOPG films, and the lowest for DPPC. The exclusion surface pressure (π_e_) is the highest π value at which the peptide is able to penetrate the monolayer. This value was higher for the DOPC/DOPG and DPPC/DOPG mixtures than for the corresponding pure lipids, pointing to a synergistic effect of the lipids in the binary systems. For these two mixtures, and also for pure DOPG, π_e_ was higher than 30 mN/m, where the lipid density is proposed to be comparable to that in bilayers [34,35,36]. Therefore, insertion of the peptide in bilayers of these compositions is likely to occur. Pure DPPC and DOPC films depicted lower π_e_ as expected, since electrostatics is known to enhance CPP incorporation [37,38,39]. The insertion occurred up to higher π_e_ values in DOPC than in DPPC, indicating that KR_9_C incorporation is more favorable in the fluid than in more rigid membranes. 

∆π depends on the amount of peptide incorporated, on the rearrangement caused by the peptide on the surrounding lipids, and on the stiffness of the monolayer (i.e., the surface pressure response of the film, due to a change in the area per lipid). In order to better compare the effect of peptide incorporation into DOPC/DOPG and DPPC/DOPG films (which depict different stiffness), we estimated the change in the mean molecular area occupied by the lipids not affected by the peptide upon incorporation of the peptide, ∆A. This was calculated with the mean molecular areas obtained from the compression isotherms of the pure lipids at π_o_ and at π_f_, as detailed in Appendix A (Appendix A) [16]. Figure 1b shows the percentage of area change, ∆A/A_0_% (A_0_ is the mean molecular area of the lipids at π_o_) as a function of π_0._ The open symbols were calculated with the data of Δπ extrapolated from the linear fitting in Figure 1a. Changes in the area observed in monolayers of DOPC/DOPG at high compactions were larger than in DPPC/DOPG monolayers, and thus, a larger amount of KR_9_C incorporated, and/or a larger perturbation was observed in these fluid films, compared to the more rigid DPPC/DOPG films.

The interaction of the peptide with monolayers was also followed by fluorescence microscopy in order to check the presence of coexisting phases. In these experiments, the lipid mixtures contained 0.5% of the Rho-PE fluorescent probe. The monolayers of the three compositions were first observed during compression, being DOPC/DOPG and DOPC/CHOL/DOPG homogeneous in the whole range, whilst phase coexistence was detected in DPPC/DOPG films at surface pressures higher than 27 mN/m. This is a consequence of the DPPC phase transition from liquid-expanded to liquid-condensed, as can be inferred from the compression isotherm, shown in Appendix A (Appendix A). The phase transition shifted from 9.5 mN/m for pure DPPC to 27 mN/m for the 1:1 mixture with DOPG.

In order to study the effect of KR_9_C on the film’s texture, monolayers with Rho-PE were formed at an initial surface pressure of 30 mN/m, and the peptide was injected in the subphase at a final concentration of 10 µM. Films of DOPC/DOPG and DOPC/CHOL/DOPG remained homogeneous after peptide addition (data not shown), whilst in DPPC/DOPG monolayers, the domains increased in size and number (see Figure 2), with a kinetic similar to that of the change in surface pressure (~30 min, data not shown). At 30 mN/m, the presence of the peptide increased the surface pressure in about 1 mN/m (see Figure 1a). This increase would lead to a negligible increase in the percent of the area occupied by the liquid-condensed domains, and not to the 25% increase, shown in Figure 2b. Thus, the increment in the area occupied by the lipids in the liquid-condensed phase was induced by the peptide, which promoted the presence of compact regions in the monolayer where the fluorescent probe was not soluble. Given that KR_9_C incorporated preferentially into less rigid monolayers (compare the changes in surface pressure-induced in DOPC and in DPPC in Figure 1a), it is not expected that the dark regions correspond to regions enriched in the peptide. Considering that KR_9_C incorporation was more favorable into DOPG than into DPPC films (Figure 1a), we hypothesize that the peptide recruited DOPG lipids, and consequently the film regions without peptide became depleted of this lipid. As a result, the regions of the monolayer where the peptide did not incorporate became enriched in DPPC. This enrichment would lead to a shift of the surface pressure for phase transition towards that of pure DPPC, thereby stabilizing the liquid-expanded phase at 30 mN/m.

### 3.2. KR_9_C Increases the Conductivity of DPPC/DOPG and DOPC/DOPG Membranes, without Affecting Those with Cholesterol

The absence of CPP insertion into the monolayers containing CHOL, together with previous reports performed with similar peptides [19,20] suggests that KR_9_C is unable to cross through membranes with CHOL. However, monolayers are model membranes with a single leaflet, thus, lacking the geometry for translocation that is present in bilayers submerged into two aqueous milieus. Therefore, we measured the change in electrical conductivity of the membranes induced by the peptide in planar free-standing bilayers composed of DOPG/DOPC, DPPC/DOPG and DOPC/CHOL/DOPG. For this, we used BLMs separating two aqueous compartments. The BLMs were subjected to a potential of 150 mV, and the current was registered before and after the addition of 1 µM peptide. Representative experiments are shown in Appendix A. The change in membrane conductivity induced by the peptide is plotted in Figure 3. As can be observed, the conductivity increase in DOPG/DOPC membranes was similar to that in DPPC/DOPG, and 3-times lower in the presence of CHOL. From these results, we conclude that CHOL not only hinders peptide incorporation into a hemilayer, according to the observations reported in the previous section, but also precludes peptide translocation.

In the case of DPPC/DOPG membranes, the increase in conductivity induced by the peptide may be explained not only considering peptide translocation, but also due to an increase in membrane defects as a consequence of the stabilization of the phase coexistence induced by the peptide. This kind of defects has been proposed to be present during phase transitions [40,41].

### 3.3. KR9C Adsorbs at the Surfaces of the Three Membrane Compositions

The results shown up to now suggest that KR_9_C did not incorporate or translocate membranes with CHOL. However, adsorption at the interface may still occur, especially considering that 50% of the lipids correspond to anionic species. In order to test the ability of the peptide to adsorb on the membrane surfaces, 100 nm-sized LUVs were exposed to increasing amounts of the peptide, and the potential at the slipping plane (z-potential, ζ) was determined. At all the tested peptide concentrations, LUVs remained intact as detected by Dynamic Light Scattering (see Appendix A). As expected, negative values of ζ were determined in the absence of peptide for the three membrane compositions, (Figure 4). As the ratio of peptide to lipid increased, ζ values became less negative, indicating peptide adsorption. 

Table 1 lists the total change in ζ (Δζ = ζ(P/L = 0) − ζ(P/L = saturation)), and the P/L value at which half of the change in ζ occurred (P/L_50_) for each membrane composition. For DOPC/DOPG and DPPC/DOPG compositions, partial neutralization was observed since ζ changed from −44 to 15 mV and from −64 to 8 mV respectively. For DOPC/CHOL/DOPG, ζ changed from −50 to 61 mV, and thus, vesicles were highly charged both without peptide and at saturation, though with an opposite charge. Regarding P/L_50_, it was an order of magnitude lower for DOPC/CHOL/DOPG, requiring higher concentration for DPPC/DOPG and even higher for DOPC/DOPG.

Our studies with monolayers and BLMs indicated that KR_9_C did not insert or translocate in membranes with CHOL. Thus, the increase in the ζ values is very likely due to the adsorption of the peptide, accumulating in the LUV’s surface without internalization. This explains the lower P/L value for saturation of these LUVs compared to DPPC/DOPG and DOPC/DOPG, as well as the higher change in ζ. 

### 3.4. Adsorption of 5-FAM-KR_9_C on Membranes with Cholesterol Is Faster and Occurs to Higher Extents than on DOPC/DOPG Membranes

The kinetics of adsorption of the peptide on the membranes was followed with the fluorescently marked peptide 5-FAM-KR_9_C using GUVs and confocal microscopy. GUVs contained 0.5% Rho-PE as a fluorescent probe and were between 5 and 20 µm in diameter. At least 16–20 GUVs of each composition were analyzed, and Figure 5a shows representative images. The time at which fluorescence of 5-FAM-KR_9_C was detected at the vesicle’s rim was variable, since it depended on peptide diffusion. We determined the fluorescence outside each GUV over time and normalized the fluorescence at the rim with the fluorescence outside it at each time, Γ_rim_ = rim’s fluorescence/fluorescence outside the GUV. Γ*_rim_* gives an insight of the excess peptide at or/and inside the membrane in relation to the bulk peptide concentration (possible changes in the fluorescent emission of 5-FAM, due to changes in the environment would also affect Γ_rim_). 1 or 2.5 μM peptide was added, corresponding to P/L ratios greater than 1 where ζ has already reached a plateau (Figure 4). Figure 5b,c show the typical profiles for Γ_rim_ as a function of time for GUVs of DOPC/DOPG and DOPC/CHOL/DOPG, respectively. The fluorescence increased over time, until it reached a constant value. GUVs composed of DPPC/DOPG did not reach a plateau, they were unstable upon peptide addition and exploded a minute after the peptide concentration outside the GUV reached a constant value, being the maximal Γ_rim_ value of about 2. Therefore, this composition was excluded from this set of experiments. Himeno et al. observed the formation of pores in DOPG/DPPC mixtures that was suppressed by the addition of salt, due to the screening of the electric charge of DOPG [42]. Contrary to their results, we observed that GUVs became more unstable when the peptide was added, although a partial neutralization was observed (see Figure 3). Therefore, the instability induced by the peptide in DPPC/DOPG membranes was not related to an electrostatic effect. It may be related to the increase in DPPC-rich regions, which are expected to be rigid, and thus, not able to deform easily when the peptide incorporated.

Plots as those depicted in Figure 5b,c were fitted with the exponential equation Γ_rim_ = *C*(1 − *e^−kt^* + *A*), where *A* and *C* fit parameters that give account of the fluorescence values when the peptide reached the membrane (Γ_rim_ = 1) and at long times (maximal Γ_rim_ value), respectively, and *k* is the effective constant for the kinetic of peptide accumulation at the membrane. Sharmin et al. studied the interaction of CF-R_9_ with GUVs of DOPC/DOPG (8:2) and DOPC/CHOL/DOPG (6:4:2) and proposed a kinetic model with three stages: Adsorption of the peptide in the external monolayer, transfer from the external to the internal monolayer, and desorption from the internal monolayer to the interior of the GUV [19]. With this in mind, peptide concentration at the membrane rim (and thereby, Γ_rim_ may increase, due to peptide adsorption and incorporation from the outside, and decrease, due to peptide desorption and entry to the GUV lumen. 

As already mentioned, 1 and 2.5 μM correspond to saturating peptide concentrations, and thus, no significant increase in the rate of peptide-membrane interaction with concentration is expected in this concentration range. In agreement to this, we found no significant differences between the *k* values at 1 and 2.5 μM, and thus, Table 2 shows the mean value for *k* at the two studied concentrations. Table 2 also shows the maximum intensity observed at long times. From these data, we can draw two conclusions. First, that the apparent kinetic for the peptide-membrane interaction process was slightly faster in the presence of CHOL, the distribution of the obtained values is shown in Appendix A (Appendix A). Second, that the amount of 5-FAM-KR_9_C appears to accumulate at the membrane in a greater extent in the presence of CHOL (the change in this parameter may also be affected by changes in the fluorescence emission of 5-FAM, due to a different environment when CHOL is present). This is in agreement with the results found with the z-potential measurements.

Considering the results reported in the literature and our own ones, the difference in the values of *k* between both compositions can be explained considering that the peptide penetrated into the DOPC/DOPG GUVs, whilst it did not cross the membranes with CHOL. Thus, not only adsorption, but also desorption into the GUV’s interior affected the value of *k* in DOPC/DOPG membranes. On the other hand, in DOPC/CHOL/DOPG membranes, *k* depends only on peptide adsorption on the vesicle surface. Therefore, 5-FAM-KR_9_C accumulated more efficiently and faster at the DOPC/CHOL/DOPG membrane rim. In this regard, Sharmin et al. found that the addition of 40% CHOL to DOPG/DOPC (2:8) membranes increased about 3-times the equilibrium constant for CF-R_9_-membrane interaction. Furthermore, we found that an increase in KR_9_C concentration leads to a more efficient accumulation of the peptide on membranes with CHOL than on membranes without this lipid, note that the maximal value for $\Gamma_{rim}$ increased 2.4-times in the presence of CHOL and only 1.2-times in its absence (Table 2). 

### 3.5. KR_9_C Softens DOPC/DOPG Membranes 

We further studied the effect of the peptide on the mechanical properties of the host membrane using the composition that showed adsorption and translocation without lysis, i.e., the membranes composed of DOPC/DOPG. This study was performed by analyzing the thermal shape fluctuations of the GUVs and using an active method. Figure 6a shows representative images for a GUV at different times in the absence and in the presence of KR_9_C. It is evident from these images that the shape of the GUV fluctuated more when the peptide was present. In order to quantify this observation, we determined the aspect ratio (AR) of GUVs along with all frames in a video, and statistically analyzed the standard deviation of this parameter (SDAR), as detailed in the Experimental Section. The value of SDAR gives an insight into how much AR changes in time for a particular GUV. A histogram of the values for SDAR at each peptide concentration for a set of experiments is shown in SI. The distribution of values of SDAR shifted to a higher number as the peptide concentration increased; thus, the number of GUVs with fluctuating shapes increased in the presence of the CPP. 

We next classified GUVs as “fluctuating” or “not-fluctuating” and calculated the population of fluctuating vesicles [28]. Figure 6b shows that the percentage of fluctuating GUVs increased in the presence of the CPP, indicating that the energy cost for a shape change decreased when KR_9_C was added.

To quantify the elasticity of the membrane upon undulations, we applied mechanical stress to the GUVs by means of optical tweezers. Membrane nano-tubes were generated from GUVs before and after the addition of low peptide concentrations (0.25 μM) and registered the retraction process after the trap was turned off. Figure 7a shows a temporal image sequence of the retraction process, from which nano-tube length was obtained. These data were plotted, as shown in Figure 7b, and fitted in order to obtain the characteristic time for relaxation with an equation that neglects inertial terms [43]: $L\left(t\right)=L_{0}\;exp\left(-t/\tau \right)$. Here, *L*_0_ and *L* are the initial nano-tube length and its temporal value, respectively. The values for R-square for the fitting were within 0.91 and 0.98, being lower for the experiments in the absence of peptide. The fitting was better at high nano-tube lengths than at the end of the experiment (see the two examples in Figure 7b). This could be related to bead-vesicle interactions that were not considered in the fitting model.

The inset in Figure 7b shows the results of two sets of independent experiments, where a 5-fold increase in τ was observed after peptide addition, with the retraction rate being independent of the number of generated nano-tubes and of the time after peptide addition. Both experiments, active curvature perturbation, as well as thermal fluctuation, pointed to a softening of the membrane in the presence of the peptide. Charged surfaces are known to be more rigid against undulations than neutral ones [44,45,46,47,48,49,50,51,52]. Therefore, the decrease in membrane rigidity could be explained, considering that the peptide partially neutralizes these membranes. Note that a 3-fold decrease in the absolute value of the z-potential was observed in the presence of the peptide (Figure 4). Besides, it was reported that the addition of DOPG to DOPC bilayers increases the bending rigidity of GUVs [53]. In this regard, our hypothesis that KR_9_C recruited DOPG lipids in DPPC/DOPG membranes could also be considered here, since peptide incorporation was favored in DOPG compared to DOPC monolayers (see Figure 1). The recruiting of the anionic lipid in the regions close to the peptide would lead to an enrichment of DOPC in other regions, subsequently decreasing the observed bending rigidity. Furthermore, τ gives an account of the viscosity to the elastic response of the system. In lipid bilayers, the viscosity term is proposed to be negligible [54], and thus, in the absence of the peptide, this term corresponds to the motion of the bead in the aqueous media. However, membrane viscosity may turn appreciable in the presence of the peptide, thereby affecting the value of τ.

Significant reduction in membrane bending rigidity has been reported in the presence of different AMPs using a variety of hosts membrane compositions [51]. This observation has been explained considering local membrane thinning, pore formation, higher lateral diffusion, peptide-peptide interactions, and/or inhomogeneous spontaneous curvature [55,56,57,58,59,60]. Addition of HIV-1 fusion peptide to membranes also reduced the bending rigidity, and, in this case, a decoupling of the hemi-layers induced by the peptide was proposed [61,62].

We here studied a hydrophilic CPP-like peptide that does not form stable pores in the membrane, and very probably translocate through hydrophilic pores, in analogy to the closely related peptide R_9_ [26]. Nonetheless, we cannot discard that some of the previously mentioned phenomena induced by AMPs are also happening in our experiments, yielding softer membranes. Regarding CPPs, a moderate reduction in the bending rigidity of DOPC membranes has recently been reported using molecular dynamics. The authors suggested that the CPP ability to convey water molecules into the lipid bilayer may be responsible for locally affecting the membrane bending modulus [63]. Therefore, this may be another explanation for the observed softening.

## 4. Conclusions

We investigated the interaction of KR_9_C, a CPP-like peptide, with anionic membranes of different compositions: DOPC/DOPG (1:1), DPPC/DOPG (1:1) and DOPC/CHOL/DOPG (3:2:5). Our results indicate that the cationic peptide adsorbs on the polar region of the three membranes, thereby reducing the anionic charge and even turning it positively charged. The surface charge change was higher in membranes with CHOL; also the fluorescently marked peptide showed higher fluorescence levels in these membranes. Thus, peptide accumulation on the membrane surface was maximal for the DOPC/CHOL/DOPG composition. However, the peptide did not insert into monolayers containing CHOL, nor increased conductivity of BLMs with CHOL. Therefore, we conclude that the decrease in permeability that is known to induce the presence of CHOL in membranes is able to preclude CPP translocation, despite the fact that the anionic membrane concentrates it on its surface. This is in agreement with previous results [19,20].

Monolayers composed of PC and DOPG (both, with the saturated DPPC and the unsaturated DOPC) were able to incorporate KR_9_C, even more than the pure lipids. The cut-off surface pressure was higher for DOPC/DOPG than for DPPC/DOPG monolayers, in both cases being higher than 30 mN/m, where the molecular density is similar to that in bilayers [34,35]. In DPPC/DOPG monolayers, an increase in the size and number of condensed domains was observed. This process occurred with a similar kinetic than that for the surface pressure increase (~half hour) and slower than the kinetics of adsorption of the fluorescent peptide before vesicle burst (~one minute). Thus, the peptide absorbed fast to the film, and the subsequent insertion promoted a slow reorganization of the monolayer with an associated phase transition.

This was explained proposing that the CPP recruited PG lipids, thus, enriching the rest of the monolayer in DPPC and thereby decreasing the surface pressure for the liquid-expanded to the liquid-condensed phase transition. This is expected because peptide penetration into DOPG monolayers was more favorable than into DPPC monolayers.

The conductivity increase induced by KR_9_C on DPPC/DOPG and DOPC/DOPG membranes was similar. However, we propose that in the case of the saturated PC, the increase in the conductivity is related to an increase in membrane defects instead of peptide translocation. This is proposed because DPPC/DOPG GUVs were unstable and exploded in the presence of the peptide, and also because phase coexistence was favored by the peptide in Langmuir monolayers.

The membranes composed of DOPC/DOPG showed good peptide insertion, with a significant increase in membrane conductivity and high stability upon vesicle lysis. In the presence of KR_9_C, these membranes got softer upon out-of-plane deformations. This was explained assuming that the peptide recruited DOPG, thus, leading to a DOPC enriched membrane. Besides, peptide adsorption leads to partial membrane neutralization, which may also contribute to the decrease in bending rigidity. Other explanations, such as the incorporation of water in the membrane coupled with peptide translocation, increase in membrane viscosity and local membrane thinning cannot be discarded.

In a cell membrane, a great variety of compositions coexist transiently in small patches, each one with different mechanical properties. Thus, pre-concentration of CPP may occur in a region enriched in CHOL, whilst the peptide inserts in a neighbor region depleted of CHOL. Under this point of view, the patchwork like the organization of the membrane would act synergistically enhancing peptide translocation. Regarding the softening effect promoted by the peptide on membranes, this could trigger in cells the activation/deactivation of metabolic pathways leading to changes in the lipidic composition of membranes in order to counteract the effect of the peptide.

## Figures and Tables

**Figure 1 biomolecules-09-00625-f001:**
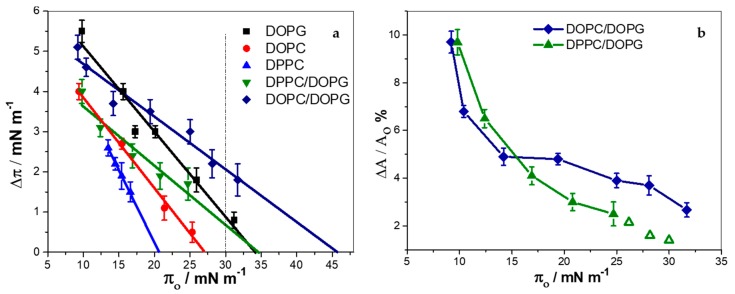
Insertion of KR_9_C into monolayers of the indicated composition (average ± SD of at least three experiments). Changes in surface pressure (**a**), or in mean molecular area (**b**) upon addition of 10 μM of peptide as a function of the initial surface pressure. The open symbols in panel (**b**) were calculated from data obtained from the linear fit in panel (**a**). π_e_ is the value of π in which Δπ = 0.

**Figure 2 biomolecules-09-00625-f002:**
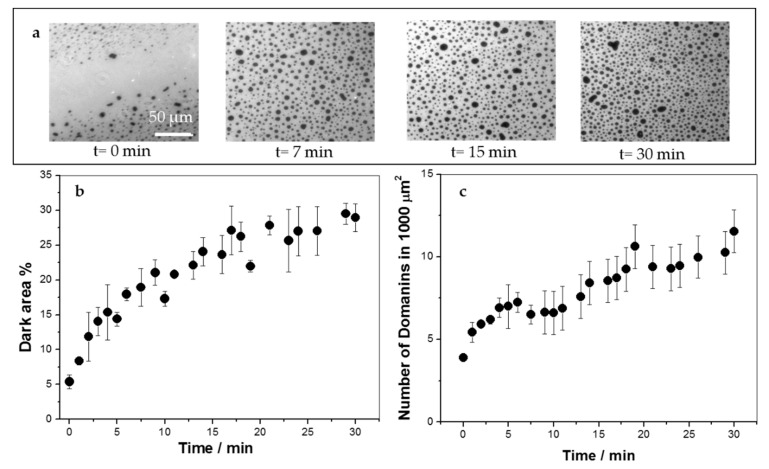
Insertion of KR_9_C into 1,2-dipalmitoyl-sn-glycero-3-phosphocholine (DPPC)/DOPG monolayers at 30 mN/m and 23 °C followed by fluorescence microscopy. Ten micrometers of the peptide were injected into the subphase at t = 0. (**a**) Representative images at the indicated times after peptide addition. (**b**) Percent of the area occupied by the darker regions, which corresponds to liquid-condensed phase state. (**c**) Number of domains in a 100 μm^2^ region.

**Figure 3 biomolecules-09-00625-f003:**
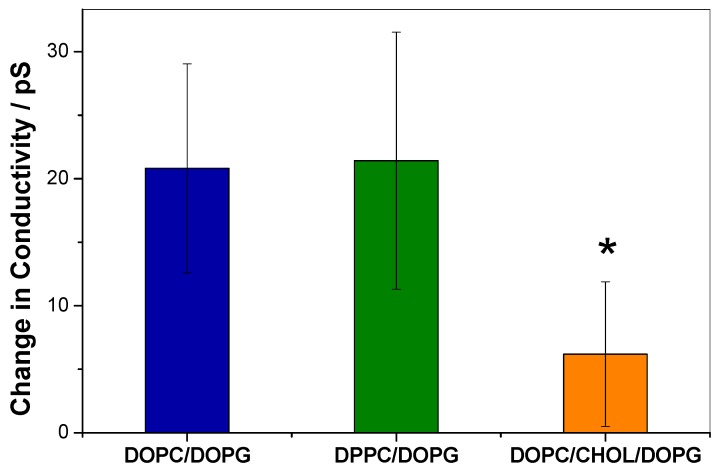
Increase in conductivity of black lipid membranes (BLMs) of the indicated compositions after addition of 1 µM KR_9_C. ANOVA test indicated that there is at least one different data with a 5% confidence (α = 0.05); the obtained * *p*-value was 0.01. The asterisk denotes a statistically significant difference between 1,2-dioleoyl-sn-glycero-3-phosphocholine (DOPC)/ cholesterol (CHOL)/DOPG and DOPC/DOPG and DPPC/DOP, according to the Tukey test

**Figure 4 biomolecules-09-00625-f004:**
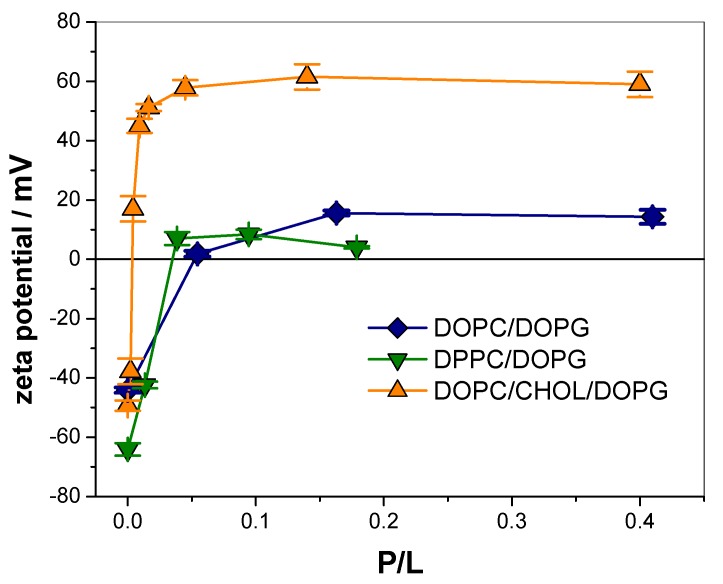
Z-potential values for LUVs of the indicated compositions incubated with increasing concentration of KR_9_C. Average ± SD of two independent experiments.

**Figure 5 biomolecules-09-00625-f005:**
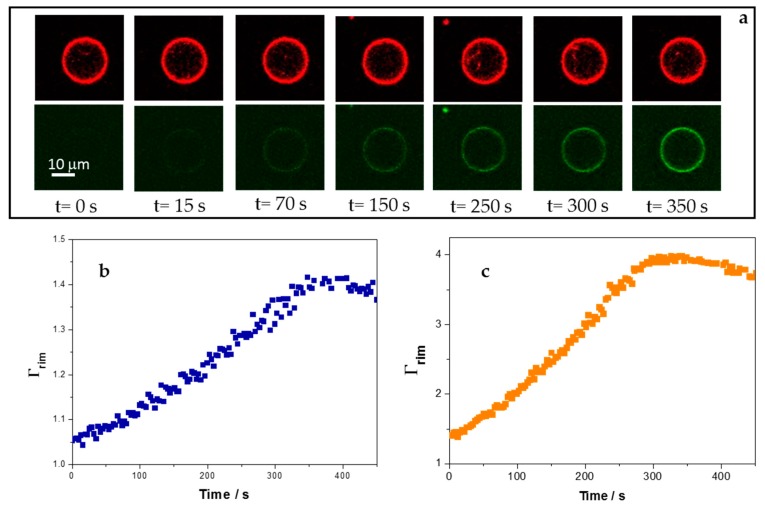
Interaction of 5-FAM-KR_9_C with giant unilamellar vesicles (GUVs) composed of DOPC/DOPG and DOPC/CHOL/DOPG. (**a**) Representative images of a single DOPC/CHOL/DOPG GUV at the indicated times after the outside fluoresce reached a constant value (2.5 µM peptide). Top panels: Rho-PE, bottom panels: 5-Figure. (**b**) Γ_rim_ = ratio of the fluorescence at the GUV’s rim and at regions outside the GUV after addition of 2.5 µM peptide for membranes of DOPC/DOPG. (**c**) Γ_rim_ after addition of 2.5 µM peptide for DOPC/CHOL/DOPG membranes. (**b**,**c**) correspond to the data of a single GUV in a representative experiment.

**Figure 6 biomolecules-09-00625-f006:**
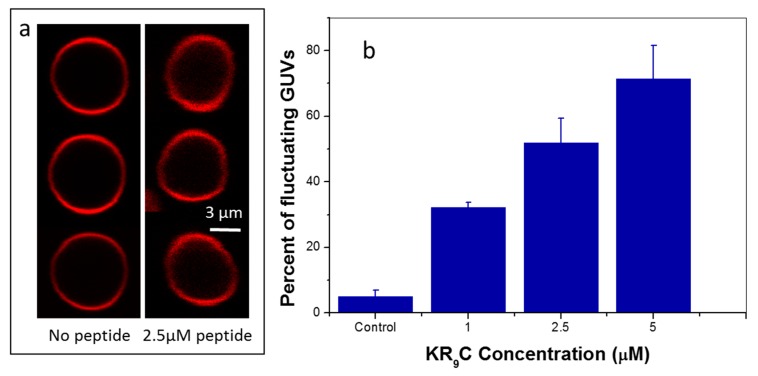
Shape fluctuations of GUVs composed of DOPC/DOPG. (**a**) Representative images. (**b**) Percent of fluctuating GUVs at each peptide concentration.

**Figure 7 biomolecules-09-00625-f007:**
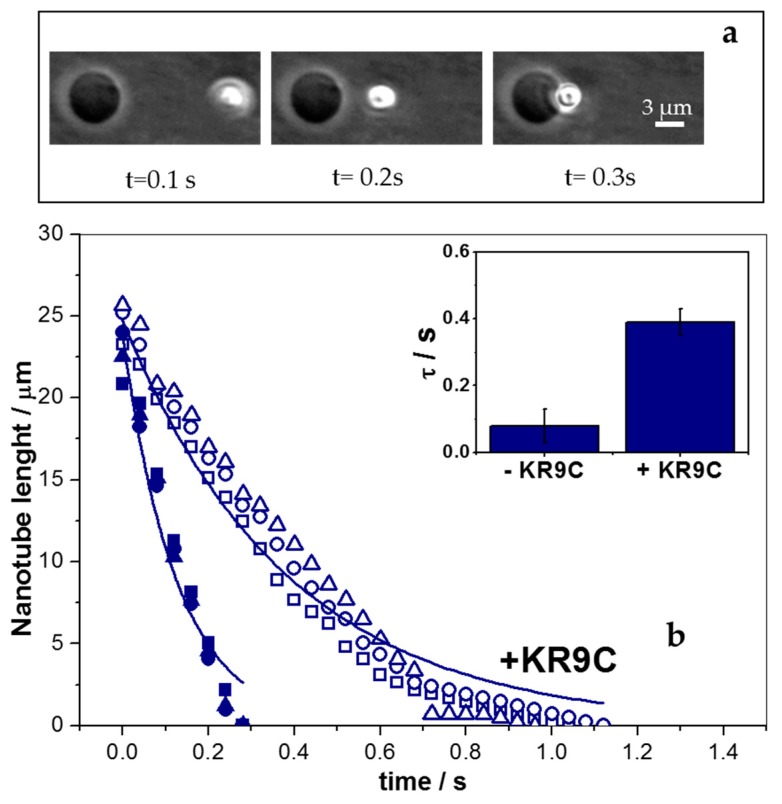
Nano-tube retraction experiments for GUVs composed of DOPC/DOPG. (**a**) Representative images at the indicated experimental times after switching off the trap in the absence of peptide. (**b**) Quantification of 3 nano-tube lengths as a function of time obtained from images, such as that in figure (**a**) in the presence (open symbols) and in the absence (closed symbols) of 2.5 μM peptide. Each symbol type corresponds to a nano-tube retraction. The two lines correspond to the fitting of the data with circles of each set of experiments. Inset: Average values for τ (±SE), obtained from two GUVs of two independent experiments.

**Table 1 biomolecules-09-00625-t001:** Data from the z-potential measurements. Total change in the ζ value (Δζ) and peptide to lipid ratio for half the total change in ζ (P/L_50_).

Membrane Composition	∆ζ (mV)	P/L_50_
DOPC/DOPG	55 ± 1	0.030 ± 0.005
DPPC/DOPG	68 ± 2	0.017 ± 0.002
DOPC/CHOL/DOPG	108 ± 3	0.003 ± 0.001

**Table 2 biomolecules-09-00625-t002:** Parameters for the interaction between the fluorescently marked peptide and GUVs. Apparent constant for the kinetics of adsorption (*k*), and maximal value of fluorescence at the rim normalized by the fluorescence outside the GUV (Γ_rim_ at long times).

Membrane Composition	*k*^1^/10^3^ s^−1^	Γ_rim_ at Long Times	
		1 µM	2.5 µM
DOPC/DOPG	3.1 ± 0.7	1.18 ± 0.07	1.4 ± 0.1
DOPC/CHOL/DOPG	4.9 ± 0.9	1.7 ± 0.3	4 ± 1

^1^ Mean ± SD of the average values at 1 and 2.5 µM peptide concentrations.

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
