# Peer review of "Interaction of a Polyarginine Peptide with Membranes of Different Mechanical Properties"

_biomolecules, 2019, doi:10.3390/biom9100625_

Round 1

Reviewer 1 Report

My comments are summarized in the attached file.

Reviewer 2 Report

The authors analyzed the significance of membrane lipid compositions for the recruitment to, penetration through, and translocation across membranes by a positively charged peptide, pointing out that the positively charged peptide is favorably targeted to a membrane region harboring cholesterol and that the peptide is translocated more readily through the membrane region that contains DOPC/DOPG. Overall, this manuscript provides novel insights into the relevant relationship between membrane composition and peptide dynamics at the contact zone, which warrants publication if the authors clearly point out scientific problems and their working hypothesis in the introduction section. Further, it is suggested that the authors provide a section of statistical analysis summarizing their methods. There are a number of minor issues indicated below, which should be taken care of before accepted into the journal.

Page 1

Lines 23-25: this sentence is not fully describing the goal of testing the basic peptide used in the project. At least this sentence should contain “peptide:.

Lines 41 and 42: regulates lateral diffusion of speciesè what species are meant by here? Are they phospholipids? Or a spectrum of membrane associated biomolecules including lipids.

Line 43: “globally”-looks not well fit here. “performed in”è performed with…

Line 53: in what extentè to what extent.

Line 60: As soè As such?

Line 62: the discover of è the discovery of.

Line 63: remove “ability”

Line 72: associate effectè the associated effects

Line 81: troughè through?.

Line72-90: a clear hypothesis set up by the authors would benefit this section. Currently this section has a tone of methods and results.

Line 101: chloroform:methanol 2:1 è in 2:1 ratio of chloroform:methanol

Line 118: t 25±1ºC. èremove underline at °C.

Line 120: PE-Rho or Rho-PE è consistency.

Line 122: Large unilamellar vesiclesè the content of this section explains the formation of multilamellar vesicles, not LUV.

Line 125: during 1 hè for  1 h.

Line 136: consistency in using unit right after numbers (50 ° C è 50 °C, 1 hourè 1 h).

Line 148: 300mM è 300 mM

Line 173: isè change into past tense.

Line 210-211: “formed by DPPC, DOPC or 211 DOPC/CHOL and DOPG”è is confusing. May need a revision.

Figure 1: Both A and B requires either standard deviation values or standard error values indicated.

In the method section it requires a section of statistical analysis.

Figure 3: provide p specific value. Was it one-way ANOVA? Or Dunnet test? can use * to indicate statistical significance.

Figurer 4: error bars are not clear. Blue diamonds are indicated with errors.

Figure 5: indicate if this is a representation of multiple experiments?

Figures 6 and 7: statistical analysis to be done.

Lines 469-471: the sentence is not clear, requiring a revision.

Reviewer 3 Report

The comments are uploaded in the separate file.

Author Response

In the manuscript by M. A. Crosio and colleagues different aspects of interaction of cell-penetrating peptide KR9C are revealed. The study combines several experimental approaches, complementing and controlling each other. The conclusions made by the authors look solid and well justified. I have only several comments on the presentation of the results and description of the experiments.

Line 129. “a 100 μm pore filter”. Probably, here should stay “a 100 nm pore filter”

Answer: this has been changed, thank you for the correction.

Lines 122-129. How zeta-potential of LUVs was measured? Which device was used? What was the composition of the background solution?

Answer: all the information is now in Materials and Methods (p.3, lines 142-147)

Lines 189 and 199. For 150 mM NaCl bathing solution, the transmembrane voltage of 150 mV seems to be high enough to induce electroporation. In analogous experiments, we determined the average waiting time of electrical breakdown of DOPC membrane to be tens of seconds at 150 mV. Strong fluctuations of the conductance illustrated in Fig. S3 before addition of the peptide may point to that transient pores already formed at such voltage. Thus, it seems that the authors studied electroporation of BLM in the presence of the peptide, rather than peptide-induced pore formation at negligible working voltage. However, qualitatively, this does not cancel results on relative change of BLM conductivity in the presence of the peptide, illustrated by Fig. 3. Please, comment on why the chosen measuring voltage was so high.

Answer: This high potential was used in order to obtain a high signal-to-noise data, since the current increases linearly with the applied potential for low potentials. For DOPC/DPPC bilayers in 150 mM salt, a linear current-voltage relationship was reported for potentials up to150 mV, becoming non-linear at higher voltages, and thus indicating that higher voltages alter the state of the membrane (K. Wodzinska, A. Blicher, T. Heimburg, The thermodynamics of lipid ion channel formation in the absence and presence of anesthetics. BLM experiments and simulations, Soft Matter 5 (2009) 3319–3330). Following that report, we chose to apply 150 mV.

Bilayers did not break in the experiments performed in this work. In previous experiments, we observed breakdown and in these cases, an abrupt current overload was observed. This did not happen in the experiments shown here.

The level of noise at 150 mV was similar to that at 0V and is related with the noise of our experimental set-up and very low level of currents that we determined (pA). The reviewer´s comment lead as to deeper analyze the signal in our experiments before and after applying the potential. We determined the dispersion of the data for the current at 0V and at 150 mV, and found that the current fluctuation was of the order of the 0.1 pN in both cases for the three membrane composition, suggesting no effect of the potential in the membrane structure. We added a comment about this in the manuscript (p. 6, line 229-232)

Lines 279-283, Figure 2. Is it possible to extract from the Figure 2 kinetics of which elementary process is the limiting one (incorporation of CPP into the monolayer; recruiting of DOPG by the peptides; phase separation in the monolayer, etc.)? Please, comment.

Answer: Thank you for this comment. We now compared the kinetic of the change in surface pressure after peptide addition with the kinetic of domain growth, and found that they are similar. We also analyzed the increase in fluorescente at the GUV´s rim before vesicle exploded, and fount that after the fluorescence outside the GUV reached a constant value, Grim increased fast (70 s) to a value double than that in the outside.

The change in surface pressure depends on both, the kinetic of peptide penetration and on the kinetic of membrane reorganization. The fluorescence at the rim indicated peptide accumulation in the membrane and in the vesicle interior. Therefore, we conclude that peptide adsorption occurs fast, and the subsequent peptide insertion induces a slow reorganization of the monolayer, leading to phase transition.

We added a phrase about this in the Conclusions (p. 15, lines 516-520)

Lines 286-288. “The absence of CPP insertion into the monolayers containing CHOL, together with previous reports performed with similar peptides [19,20] suggests that KR9C in unable to cross liquid-ordered membranes.” DOPC:CHOL membranes are liquid-disordered, not liquid-ordered.

Answer: This depends on the proportion of cholesterol. In a previous work we studied the fluidity and the permeability of DOPC/CHOL (8:2) and found properties compatible with liquid-ordered phase state (Mangiarotti et al. BBA-Biomem 2019). Furthermore, Smith et al. studied the phase diagram of ternary mixtures of DOPC/SPM/CHOL, they proposed a uniform Lo phase in a region of low SPM content, without indicating a lower limit in SPM (A.K. Smith, J.H. Freed, Determination of tie-line fields for coexisting lipid phases: An ESR study, J. Phys. Chem. B. 113 (2009) 3957–3971. doi: 10.1021/jp808412x).

As far as I know this binary mixture has not been studied deeply enough in order to ensure Lo or Ld phase state. We therefore modified this phrase and erased the term “liquid-ordered”.

Lines 319-320. Figure 4. How long was the incubation time of LUVs with the peptide before measurements of the zeta-potential? Was this time sufficient to reach the equilibrium of LUV internal volume and bathing solution with respect to the peptide?

Answer: The incubation time was 1 h (it was enough for reaching equilibrium). The information is now in the experimental section (p. 4 line 144).

Lines 343-344. “We determined the fluorescence outside each GUV over time and normalized the fluorescence at the rim with the fluorescence outside it at each time”

Where exactly was determined the fluorescence outside each GUV, at which distance from the membrane? Was the outside fluorescence spatially homogeneous?

Answer: the gray level was determined in different regions outside each GUV and close to it, and the data taken form 3-5 ROIs in different regions were averaged. We now added an example in SI.

Lines 345-346. “G??? is a measure of the excess peptide at or/and inside the membrane in relation to the bulk peptide concentration.” 2

This is the case only if the fluorescent label intensity is independent on its environment, especially, on its polarity/hydrophobicity. Otherwise, the G??? is combined characteristics of the peptide excess and spectral properties of the label. Please, comment.

Lines 369-371. “Second, that the amount of 5-FAM-KR9C accumulated at the membrane was higher in the presence of CHOL.”; Lines 380-383.

The same concern. This is the case only if cholesterol does not directly influence the intensity of the fluorescent label. Please, justify.

Answer: We agree with referee´s comment, the fluorescent spectra of the probe may change and thus, the fluorescence ratio is not exactly equal to an excess. This fluorescent probe(5-carboxifluorescein) is hydrophilic, and thus, we expect that it is positioned in the aqueous region of the bilayer and not inside the bilayer. Thus, we do not expect an important change in the spectra. However, since we cannot assure this, we changed this part of the manuscript considering this possibility (p. 10, lines 376-377 and p. 11, lines 409-410)

Lines 422-423. The equation for L(t) looks strange as dimensional quantity (–t/) stays in the exponent. On the first look, the dependences in Fig. 7b do not resemble exponential ones; please, provide the fitting curves for the data points. The negligible influence of the viscous flow of the membrane tether on the retraction kinetics should be justified. Otherwise, the characteristic time  will give information on combination of changes of membrane elasticity and its viscosity; it is not obvious how to formally separate these contributions. The viscosity may increase due to recruitment and ordering of charged lipids by the peptide.

Answer: thank you very much for this observation, the equation of the old version of the manuscript was wrong, we did not use that equation for the fitting. We now corrected this part of the manuscript (p. 13, lines 463-499). The data obtained from fitting is the characteristic time of relaxation, which includes both, elastic and viscous terms. The viscous term is that of the bead on the aqueous environment plus some possible term related with the viscosity of the membrane, which in lipid bilayers has been shown to be negligible (Derenyi et al. Ref 52). As the reviewer indicated, this could change in the presence of the peptide. We now included this possibility in the discussion (p. 14, lines 491-494). Thank you very much for the observation.

Line 635, reference [47]. “Ermakov” should stay instead of “Ermakiv”.

Answer: This was corrected.